# The Prevalence of Injuries and Traumas in Elite Goalball Players

**DOI:** 10.3390/ijerph17072496

**Published:** 2020-04-06

**Authors:** Anna Zwierzchowska, Barbara Rosołek, Diana Celebańska, Krystyna Gawlik, Martyna Wójcik

**Affiliations:** 1Institute of Sport Science, The Jerzy Kukuczka Academy of Physical Education in Katowice, 40-065 Katowice, Poland; a.zwierzchowska@awf.katowice.pl; 2Department of Physical Education and Adapted Physical Activity, The Jerzy Kukuczka Academy of Physical Education in Katowice, 40-065 Katowice, Poland; d.celebanska@awf.katowice.pl; 3Physiotherapy Department, The Pope John Paul II State School of Higher Education in Biala Podlaska, 21-500 Biała Podlaska, Poland; k.m.gawlik@gmail.com; 4Student Research Group of Adapted Physical Activity, The Jerzy Kukuczka Academy of Physical Education in Katowice, 40-065 Katowice, Poland; martyna242424@gmail.com

**Keywords:** training, Paralympic, parasport, injury, goalball

## Abstract

Background: The last decade has seen the dynamic development of Paralympic sport, including the development of training facilities and training methods that allow for the achievement of results at the highest level in this group. This may be associated with an increased risk of injury and traumas. This study aimed to evaluate the prevalence and locations of injuries and the types of trauma suffered by goalball players. Methods: The study covered 43 players (24 women and 19 men) of the Goalball European Championship. A questionnaire survey was conducted. Results: The injuries were reported by 44%. Most often they concerned the upper limbs (92%). Positive correlations were found between age and prevalence of pelvis and knee joint injuries, whereas negative correlations occurred between competitive experience and the prevalence and location of injuries in the area of the upper limb. The highest percentage of trauma was found for epidermal abrasions, contusions, and dislocations. A negative correlation was observed between age and the prevalence of epidermal abrasions, whereas a positive correlation occurred between age and the prevalence of sprains and dislocations. The competitive experience was negatively correlated with the prevalence of abrasions and contusions and positively correlated with the prevalence of sprains and dislocations. A statistically significant correlation was found between body mass (BM) and the prevalence of injuries. In 23% of cases, training was interrupted for more than one month, whereas in 43%, the break was below one month. Conclusions: Playing position in goalball does not affect the prevalence of injuries and traumas, while body mass has a moderate effect on the prevalence of these events. The age of the subjects and their sports experience impact significantly on the prevalence and types of injuries.

## 1. Introduction

The last decade has seen the dynamic development of Paralympic sport [1], including the development of training facilities and training methods that allow for the achievement of results at the highest level in this group [2,3,4]. These factors may be associated with an increased risk of injury and traumas [5]. The identification of these problems is important for the health of athletes with disabilities, but also for the planning of prevention strategies. Goalball is one of the Paralympic team sports for athletes with vision impairment [4]. Two teams (both of three athletes) compete to score goals by throwing or rolling a ball that produces sound when in motion. Players remain in their area in both defence and attack (there is no contact with opponents). In offence, the player takes several steps or makes the rotation of the body before a throw. Defence players lie on their hips, stretch arms above heads and extend legs to cover as much area as possible [2,3,4]. According to the regulations of the International Blind Sports Federation (IBSA) [4], all competitors must wear eyeshades (which completely cover the eyes) whereas other pieces of clothing may be equipped with protectors that do not protrude from the body by more than 10 cm. It is emphasized that athletes with disabilities are more at risk of injury than nondisabled athletes [6], and goalball is a high-risk sport [7,8]. Willick et al. [9] showed that during the Paralympic Games in London (2012), goalball was the third among sports with the highest incidence rate (IR = 19.5). Therefore, we found it appropriate to address this issue in our research. The study aimed to evaluate the prevalence and locations of injuries and the type of traumas to goalball players and to identify their relationship with age and competitive experience. It was assumed that the training experience, age, position, and somatic parameters of the athletes are important factors for the prevalence of injuries in this group.

## 2. Materials and Methods 

### 2.1. Material

The study covered 43 players (24 women (W) and 19 men (M)) of the Goalball European Championship, Division B, from Spain (*n* = 6), France (*n* = 3), Israel (*n* = 3), Finland (*n* = 2), Slovenia (*n* = 1), Ukraine (*n* = 2), Hungary (*n* = 4), Portugal (*n* = 2) Great Britain (*n* = 2), Netherlands (*n* = 3) Greece (*n* = 1), and Poland (*n* = 14). Characterization of the study participants is presented in Table 1. 

The athletes were qualified according to the criteria of the International Blind Sports Federation [1] to groups B1 (visual acuity less than LogMAR 2.60), B2 (visual acuity ranges from LogMAR 1.50 to 2.60 (inclusive) and/or the visual field is constricted to a diameter of fewer than 10 degrees), and B3 (visual acuity ranges from LogMAR 1 to 1.40 (inclusive) and/or the visual field is constricted to a diameter of fewer than 40 degrees). Twenty five players (58%) were classified in group B3, 15 (35%) in B2, and 3 (7%) in B1. All athletes declared that during the training they were using the kits used in the competitions, equipped with protective equipment that is compliant with IBSA regulations [4]. Mean age of the study participants was 26 years, whereas the mean training experience was 6 years. Most players (63%) declared two to three training sessions a week, 28% declared four and more training sessions, whereas 9% reported one training session a week. Six players played as centers (14%), 29 (67%) as wings, and 8 players (19%) played in both positions. 

### 2.2. Methods

A survey questionnaire was used in the study, concerning the prevalence and sites of injuries and the types of trauma. Following the definition of the International Olympic Committee (IOC), we assumed that sports injury is “damage to body tissue resulting from practising a sport or exercise” and we also used the time of absence from training and competitions as a criterion for classification of injury. However, we considered traumas to be the consequences of sporting events that had affected the musculoskeletal system or caused a concussion and had been classified as a disease manifested by general symptoms from body systems and organs [10,11,12].

The questionnaire contained items divided into three sections. The first section concerned the respondent data and information related to practising goalball. The second section concerned the prevalence of injuries (frequency, site, and pain duration). The third section concerned traumas (type, prevalence, and form of rehabilitation). The questionnaire was prepared in English. The survey was carried out during the Goalball European Championship (Division B) in Chorzów, Poland by the authors assisted by a coach and team interpreter. Participants gave their oral consent to participate in the survey. 

### 2.3. Bioethics Committee

The research is part of the project “Lifestyle of disabled players in Paralympic sports in the context of health risks”, which was approved by the Bioethics Committee of The Jerzy Kukuczka Academy of Physical Education in Katowice (No. 9/2012) (informed consent was obtained from each patient included in the study, and the study protocol conforms to the ethical guidelines of the 1975 Declaration of Helsinki as reflected in a priori approval by the institution’s human research committee). 

### 2.4. Statistical Analysis

Statistical analysis was performed using the Statistica 10 software. The differentiation of somatic parameters in terms of gender was verified (Student’s t-test). The percentage of injuries in individual sites and the percentage of types of injuries were recorded, and their relationship to age, competitive experience, and somatic parameters were verified by evaluating correlations using the Pearson’s test. The prevalence of injuries differing depending on sex (women (W) vs men (M)), sports skill level, and playing position (centre vs wing) was verified. The statistical significance was set at *p* < 0.05.

## 3. Results

Injuries occurring over the competitive careers were reported by 44% (54% W, 32% M) of athletes. Sex, sports skill level, and position (centre/wing) did not significantly modify the prevalence of injuries (*p* = 0.14; *p* = 0.66; *p* = 0.73). The detailed data concerning the site of injuries are contained in Table 1. The most frequent injuries occurred to the upper limb and accounted for 92% of all reported injuries. Positive statistically significant correlations were found between age and prevalence of pelvic and knee joint injuries. Furthermore, statistically significant moderate negative correlations were observed between competitive experience and the prevalence of injuries to the upper limb area (elbow joint, forearm, hand, and wrist) (Table 2). 

The highest percentage of injuries was found for epidermal abrasions, contusions, and dislocations (Table 3). A significant negative correlation was observed between age and the prevalence of epidermal abrasions, whereas positive correlations occurred between age and the prevalence of sprains and dislocations (Table 3). Competitive experience was negatively correlated with the prevalence of abrasions and contusions and positively correlated with the prevalence of sprains and dislocations.

Furthermore, a statistically significant correlation was found between body mass (BM) and the prevalence of all injuries (injuries requiring a break of less than one month + injuries requiring a break of more than one month) (R = 0.69; *p* < 0.002), and between BM and the prevalence of injuries that required a break of more than one month (R = 0.64; *p* < 0.005). 

In 23% of cases, the training following traumas was interrupted for more than one month. Of these, 8% of traumas were surgically treated. In 42% of cases, the injury resulted in a training break below one month, while in 8% it was not necessary to have a training break. It was noted that following 23% of the injuries, the athletes underwent a rehabilitation process.

## 4. Discussion

The scientific literature has demonstrated that the identification of factors that generate injuries and/or trauma, their prevalence, their type, and their nature are important issues for sports science [6,13,14,15,16,17,18]. Only proper recognition of this problem is the basis for optimal training, without compromising the health of the athlete. However, most of the research work has been done on able-bodied athletes and the identification of factors affecting injuries and/or traumas depending on the type of sport. Among other things, a significant effect of somatic features and body composition indicators predisposing to injuries and/or trauma has been emphasized [19,20,21,22,23,24]. Our results are consistent with these findings, as body mass significantly correlated with the prevalence of injuries among goalball players. On the other hand, although this sport requires a high degree of agility and quick changes of the position from high to low, no correlation was found between body height and the prevalence of injury and/or trauma. Gender was also not a factor differentiating these events, which is not consistent with studies of able-bodied athletes [19,20,21,22,23]. 

There are few studies in the literature concerning injuries and/or traumas in visually impaired athletes, especially considering the specificity of one sport. This was confirmed in studies by Fagher et al. [6] and Magno e Silva et al. [14,15,25], where only a synthetic approach to the problem was presented in terms of the prevalence of injuries among swimmers, track-and-field athletes, soccer players, and judo players. This methodological approach to the problem is understandable, because as Ferrara et al. [9] explain in their study, research in Paralympic sports concerns a relatively small number of participants, especially if the structuring of the group by type and degree of disability, the sport practised, the medical qualification for the sports group, and sex are taken into account. Such a purposeful selection leads to obtaining a heterogeneous group that is small in number, which in consequence makes it difficult to draw conclusions based on statistics. Our research confirms this thesis; we have examined 81% of all the B group European championship elite goalball players who were somatically gender-differentiated (*p* < 0.05), and while the age and frequency of injuries and/or traumas did not differentiate the respondents, there were only *n* = 43 athletes. 

Few studies have examined injuries or traumas to people with visual disabilities, let alone publications describing this problem in players practising goalball, being the only Paralympic team sport dedicated only to athletes with visual disabilities. Our research evaluated the prevalence of injuries among the players of the Goalball European Championships, mainly in the area of the upper limbs, as observed by Gajardo et al. [21]. However, detailed results in this area are not fully consistent with those presented by Gajardo [21], who observed the most shoulder or clavicle injuries in the area of the upper limb, which was not confirmed in our study, where the fingers were the most frequent sites of injuries, regardless of the age and experience of the competitors (no correlation for these variables). It is logical that in sports where catching and throwing are the basic principles, the shoulder girdle, arms, and palms of the hands are particularly prone to injuries. It is difficult to explain unequivocally the discrepancy of the results between Gajardo et al. [21] and our results. However, it should be noted that the mean age of respondents in the study by Gajardo et al. was 41 years (sd = 14.96; min–max = 15.4−71 years), whereas in our study of elite goalball players, the mean age was 26 years (sd = 7.1) (min–max.= 17−49 years), and this might have been the cause of more prevalent shoulder or clavicle injuries in the study of Gajardo. This was similar to the prevalence during the Paralympic Games in London (2012), where the most injuries were reported in shoulders, but the results additionally concerned all athletes participating in the Paralympics [9]. 

Observations of other authors conducted among blind soccer players, swimmers, and athletes showed that the location of injuries is closely related to the sport and playing technique [14,15,25]. During defensive playing and in attack, the upper limbs are exposed to heavy loads. A player performing defensive elements usually adopts an intermediate or low position, with the lower limbs in direct contact with the ground and the arms representing a support point for the upper body segments. The target defensive position is to lie on the side with straightened and stiffened lower and upper limbs with the head between the arms. There was no correlation between the prevalence of upper limb injuries and age, but this (moderate) relationship was observed in relation to the competitive experience. Among the respondents with shorter competitive experience, injuries to the elbow joint, forearm, wrist, and hand were more frequent. 

Furthermore, players with more competitive experience, and consequently a higher sports skill level and level of physical fitness, showed predictability of behaviour, which resulted in lower injury rates to these body segments. There was also a moderate correlation between the prevalence of knee injuries and the age of competitors, and a strong correlation between the prevalence of pelvic injuries and age. Older players were more often prone to injuries to these body segments. Most of the injuries suffered by blind athletes (76%) were mild (abrasions and contusions), which was also observed by Magno e Silva [5] in a synthetic analysis of injuries to blind athletes, soccer players, judokas, swimmers, and goalball players. 

Searching for the correlation between the prevalence of traumas caused by injuries with age revealed that minor traumas were characteristic of younger players, while those more severe (sprains and dislocations) were more frequent in older ones.

A similar pattern was observed for the relationship between competitive experience and the type of injury. Players with shorter competitive experience were often moderately more likely to experience abrasions and contusions, while older athletes suffered more often from sprains and dislocations.

One limitation of this study is the small number of respondents. However, the respondents were goalball players, which is a sport that is practised in the world only by blind and visually impaired people, which makes it highly specific. Furthermore, the athletes studied were a group selected from the best players in this sport.

## 5. Conclusions

Playing position, sex, and sports skill level in goalball does not affect the prevalence of injuries and traumas. The age of the subjects and their sports experience impact significantly on the prevalence and type of injuries and traumas. Older players are more likely to suffer injuries that require treatment, while the greater the sports experience, the fewer minor sporting injuries. Players participating in the Goalball European Championship (Division B) mostly suffered injuries in the area of the upper limb, especially the fingers. Furthermore, we demonstrated that body mass has a moderate effect on the prevalence of injuries.

In conclusion, the results of the examinations of elite athletes with visual impairments conducted during the Goalball European Championship (Division B), which was held in Chorzów, Poland, come from one of the few studies, in addition to the findings of Gajardo [21], that concern injuries and traumas in elite goalball players. The results are the basis for developing preventive strategies in training and during competitions to provide support to reduce the risk of injury or traumas in this group of athletes. 

## Figures and Tables

**Table 1 ijerph-17-02496-t001:** Characterization of study participants (BM—body mass, BH—body height, and BMI—body mass index).

Characteristic and Index	Women (*n* = 24)			Men (*n* = 19)			*p*-Value
	$\bar{X}$	sd	Min–Max	$\bar{X}$	sd	Min–Max	
Age	27	7.4	18–49	24	6.3	17–38	-
BM	64.9	10.7	43–91	78.2	14.4	51–100	0.001
BH	1.66	9.1	153–188	1.75	11.6	150–198	0.007
BMI	23.3	3	14.0–28.0	25.3	4.4	18.1–36.3	-

sd: standard deviation.

**Table 2 ijerph-17-02496-t002:** Site of injuries vs. age and competitive experience.

Site	Injury Rate [%]	Age	*p*-Value	Competitive Experience	*p*-Value
		Pearson’s r		Pearson’s r	
shoulder joint	2	-	-	-	-
elbow join	17	-	-	R = (−0.6)	0.005
forearm	17	-	-	R = (−0.6)	0.005
wrist	18	-	-	R = (−0.5)	0.019
hand	17	-	-	R = (−0.6)	0.005
fingers	22	-	-	-	-
foot	2	-	-	-	-
pelvis	2	R = 0.7	0.001	-	-
knee joint	5	R = 0.5	0.029	-	-
foot	2	-	-	-	-

**Table 3 ijerph-17-02496-t003:** Traumas vs. age and competitive experience.

Traumas	Traumas Rate [%]	Age	*p*-Value	Competitive Experience	*p*-Value
		Pearson’s r		Pearson’s r	
skin abrasion	39	R = (−0.5)	0.04	R = (−0.5)	0.028
contusion	37	-	-	R = (−0.6)	0.011
sprain	2	R = 0.7	0.001	R = 0.7	0.000
overload	2	-	-	-	-
dislocation	12	R= 0.5	0.047	R = 0.6	0.006
bone fracture	5	-	-	-	-
other	2	-	-	-	-

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
