# Peer review of "The Prevalence of Injuries and Traumas in Elite Goalball Players"

_ijerph, 2020, doi:10.3390/ijerph17072496_

Round 1
Reviewer 1 Report
Thank you for the opportunity to review your manuscript.
The topic of the study entitled “The prevalence of injuries and traumas in elite goalball players” reported by Zwierzchowska et al. represent a relevant and important area of research. The number of published manuscripts on this topic (goalball injuries is fairly limited) is still limited and, therefore, it is of great interest to increase the knowledge about this research subject.
In this present form this paper is not worthy to be published in IJERPH. The manuscript is incomplete, many references are lacking and some parts are unclear and poorly presented. An in depth-revision in needed before re-submission of the manuscript to this journal or to other one.
I hope that my comments will be helpful for the improvement of the manuscript. In my opinion the following remarks should be considered:
Introduction
In general this section is unclear and incomplete. The introduction should be better structured following a logic and progressive order.
In lines 34-49, please include information about the kind of clothing and protections that was used by the athletes. Authors should also reference previous investigations about the influence of the sport protections used on the injury incidence.
It would also be interesting to include some epidemiological data related to injuries in Goalball. Here you have some examples:
- Rudolph, L., & Willick, S. E. (2017). Review of Injury Epidemiology. Adaptive Sports Medicine: A Clinical Guide, 51.
- Rudolph, L., & Willick, S. E. (2018). Review of Injury Epidemiology in Paralympic Sports. In Adaptive Sports Medicine (pp. 51-58). Springer, Cham.Gajardo, R., Aravena, C., Fontanilla, M., Barría, M., & Saavedra, C. (2019). Injuries and Illness Prevalence Prior to Competition in Goalball Players. Journal of Visual Impairment & Blindness, 113(5), 443-451.
- Willick, S. E., Webborn, N., Emery, C., Blauwet, C. A., Pit-Grosheide, P., Stomphorst, J., ... & Derman, W. (2013). The epidemiology of injuries at the London 2012 Paralympic Games. Br J Sports Med, 47(7), 426-432.
- Van Rensburg, D. C. J., Schwellnus, M., Derman, W., & Webborn, N. (2018). Illness among Paralympic athletes: epidemiology, risk markers, and preventative strategies. Physical Medicine and Rehabilitation Clinics, 29(2), 185-203.
- Ottesen, T., Mashkovskiy, E., Gentry, M., Jensen, D., Webborn, N., & Tuakli-Wosornu, Y. (2018). Acute and chronic musculoskeletal injury in para-sport: A systematic review. Annals of Physical and Rehabilitation Medicine, 61, e163.
In lines 46-47 the objective should be re-written, it should be in accordance with the results of this study. Authors report data referred to location of injuries vs. injury rate and its association with age and competitive experience. Results about types of injuries vs. age and competitive experience are also reported.
Materials and Methods
I suggest to structure the Materials and Methods section as follows: study design, Bioethics Committee, Declaration of Helsinki, sample, informed consent, inclusion criteria, exclusion criteria, survey questionnaire…
In lines 87-88, authors have to include the code of the Bioethics Committee of The Jerzy Kukuczka Academy of Physical Education in Katowice.
In lines 57-70, the figure 1 is not informative. I suggest to delete this figure and to state the characterization of study participants in the text as follow: Woman (n=34), 24±7,4 age, 64,9±10.7 kg,….
In lines 78-79, this sentence “25 players (58%) were classified in B3 group, 15 (35%) in B2 and 3 (7%) in B1” correspond to the results section not to the material and methods.
In lines 81-83, authors should reference and explain in detailed the survey questionnaire. In addition, information related to its validity and reliability.
In line 84, authors indicate that “The questionnaire was prepared in English”. Please, explain how athletes filled out the questionnaire taken into account their different nationalities and their visual impairment.
In line 94, it would be interesting to show the association between injuries and the somatic parameters of the athletes.
In lines 97-114, in order to compare the results of the present study with previous reports analysing the same sport or other sports adapted to visually impaired athletes, authors should adapt this study following the “Consensus statement on injury definitions and data collection procedures” described in Fuller, C. W., Ekstrand, J., Junge, A., Andersen, T. E., Bahr, R., Dvorak, J., ... & Meeuwisse, W. H. (2006).
According to the Consensus the main important point that should be considered are related to the definition of injury and the results about the incidence. In this sense, it is essential in the present study to define the concept of injury and the professional profile of the person that registered the injury (physician, physiotherapist, physical trainer, trainer,...). With regard to the results, authors should state the prevalence of injury in sport is the incidence per 1000 hours of competition. Also results about the absolute and relative number of injuries per body region, injury type and mechanism of injury, body site, injury type, mechanism, severe Injuries and nature of injury (acute vs overuse), moment of game, period of play, and playing position are of great importance and value in this kind of studies. Finally, authors should analize the relationship between these descriptive parameters with the age, sex, type of visual impairment.
In line 115 I think that the title of the table should be “Types of injuries vs. age and competitive experience”. Please, check.
Discusión
In lines 126-177 I suggest to discuss the present results with previous studies. Some studies regarding this topic have been named before.
Please, justify the low incidence of shoulder injury or pain in this study. It is surprising that this injury has not been found in this competition. This is in disagree with previous reports. Please, discuss this point.
- Gajardo, R., Aravena, C., Fontanilla, M., Barría, M., & Saavedra, C. (2019). Injuries and Illness Prevalence Prior to Competition in Goalball Players. Journal of Visual Impairment & Blindness, 113(5), 443-451.
- Willick, S. E., Webborn, N., Emery, C., Blauwet, C. A., Pit-Grosheide, P., Stomphorst, J., ... & Derman, W. (2013). The epidemiology of injuries at the London 2012 Paralympic Games. Br J Sports Med, 47(7), 426-432.
Inline 178. The conclusions are unclear and not well structured. They should be in accordance with the objective. Please, revise this section and rewrite the conclusions.
Finally, I consider that authors should suggest preventive measures to reduce the incidence of skin abrasion, contusion and dislocation in upper limb, which are the injuries more frequent in this study.
What do you want to do ? New mailCopy What do you want to do ? New mailCopy What do you want to do ? New mailCopyAuthor Response
Please see the attachment.

Reviewer 2 Report
Good work presented, in an interesting topic, but several changes and improvements are needed in the manuscript.
Please, the authors should rewrite the abstract. For example, the main aim is presented as background, results are missing some data, and the conclusion is missing.
The introduction should be improved. There is a lack of previous investigation, and this compromises understanding the relevance of the purpose of the study. The rationale should be improved.
The Methods section misses some relevant information. Figure 1 is perhaps better understood if presented in a Table. More details and specific information about the implementation of the survey should be presented, as there could have been some BIAS issues regarding the influence of context (researcher, questions, voice, other opinions, emotional state...and so on).
Regarding statistics, why did you choose the parametric T-student test?
In the results, please provide the specific value of significance for each R provided. Please provide the values of all the relationships.
Discussion should be divided into paragraphs for clear reading.
Was there any reason for the relationship between competitive experience and contusion? Anything related to the participants' trust and will?
Please provide a paragraph on the limitations of the study, as there are a few.
Round 2
Reviewer 2 Report
Congratulations to the authors on the changes that were performed, improving the quality of the manuscript.